# Global earthworm distribution and activity windows based on soil hydromechanical constraints

Siul A. Ruiz [1,2,5✉], Samuel Bickel [1,3,5] & Dani Or [1,4]

Earthworm activity modifies soil structure and promotes important hydrological ecosystem functions for agricultural systems. Earthworms use their flexible hydroskeleton to burrow and expand biopores. Hence, their activity is constrained by soil hydromechanical conditions that permit deformation at earthworm's maximal hydroskeletal pressure ($\approx$200kPa). A mechanistic biophysical model is developed here to link the biomechanical limits of earthworm burrowing with soil moisture and texture to predict soil conditions that permit bioturbation across biomes. We include additional constraints that exclude earthworm activity such as freezing temperatures, low soil pH, and high sand content to develop the first predictive global map of earthworm habitats in good agreement with observed earthworm occurrence patterns. Earthworm activity is strongly constrained by seasonal dynamics that vary across latitudes largely due to soil hydromechanical status. The mechanistic model delineates the potential for earthworm migration via connectivity of hospitable sites and highlights regions sensitive to climate.

[1] Institute of Biogeochemistry and Pollutant Dynamics, Soil and Terrestrial Environmental Physics, Swiss Federal Institute of Technology (ETH Zürich), Zürich, Switzerland. [2] Faculty of Engineering and Physical Sciences, Bioengineering Group, University of Southampton, Southampton, UK. [3] Institute of Terrestrial Ecosystems, Physics of Soils and Terrestrial Ecosystems, Swiss Federal Institute of Technology (ETH Zürich), Zürich, Switzerland. [4] Division of Hydrologic Sciences, Desert Research Institute, Reno, NV, USA. [5] These authors contributed equally: Siul A. Ruiz, Samuel Bickel. ✉email: s.a.ruiz@soton.ac.uk

Subterranean activity by earthworms influences soil structure and provides numerous ecosystem services[1]. Soil biopores formed by burrowing earthworms serve as preferential pathways for water flow and aeration[2]. They are hot spots of biological activity that can be reused by growing roots, improve groundwater recharge and soil water retention, and support oxic conditions in soil profiles[3,4]. In locations with abundant plant-derived particulate organic carbon (POM), earthworms ingest POM-rich soil[5] and often line their burrows with secreted castings. Soil ingestion by earthworms can augment microbial activity and stimulates the formation of soil aggregates[6]. Overall, earthworm activity is attributed to significant enhancement in specific crop yields of up to 25%[7]. Empirical evidence suggests that earthworms are efficient "ecosystem engineers"[8] and play a prominent role in remediating adverse soil compaction[9] that affects nearly 5% of the world's arable land (about 68 Mha)[10].

Soil bioturbation by earthworms is driven by subterranean resource exploration at rates and frequencies that are linked to the availability of soil organic carbon from decomposing plant residue[2] and their mechanical ability to move in the subsurface. The soil hydromechanical conditions[11] link soil strength with soil water content and regulate earthworms' ability to burrow through soil. The kinematics of earthworm burrowing rely on locally extending the frontal segments of their body to mechanically penetrate the soil, followed by subsequent expansion of these segments to anchor and recollect extended segments, thereby pushing themselves through the soil[12,13]. The local pressures required by the earthworm hydroskeleton for expanding a new burrow are the primary determinants of penetration-cavity expansion[13] and vary widely with soil type and hydration conditions. Availability of spatially resolved soil properties and climatic records of soil hydration conditions offer opportunities for harnessing spatial and dynamic information to identify potential earthworm habitats at high resolution[14]. Ecological studies have provided insight into regional earthworm distributions[15,16] along with earthworm seasonal activity windows[17,18]. In addition to innate ecological patterns, physical constraints may affect earthworms' behaviors that include sensitivity to temperature, soil compaction, and soil moisture[19].

Physical bounds on earthworm bioturbation have been quantified recently by considering the interplay of soil hydromechanical constraints and biomechanical limit pressures that could be exerted by the earthworms' hydroskeleton[11]. These insights allow delineation of regions that permit bioturbation activity and offer a biophysical and climatic context for global earthworm abundance and distribution[14,15,20]. Mechanistic models could predict consequences of agricultural intensification with potential for soil compaction while simultaneously considering climatic shifts that would affect future earthworm bioturbation activity windows (e.g., dormancy during dry seasons in Mediterranean climates) and associated ecosystem services.

Here, we provide evidence that climatic conditions and highly dynamic soil mechanical states are the primary constraints for global earthworm occurrence and activity. The seasonal and dynamic nature of soil moisture conditions in many regions defines temporal activity windows that support bioturbation and shape biogeographic patterns[11]. The objectives of this study were: (i) to model soil hydromechanical conditions and derive temporal windows of potential earthworm burrowing activity, (ii) to delineate geographic regions where earthworm activity would be mechanically prohibited, and (iii) to compare predicted regions with earthworm presence data at the global scale.

We present a mechanistic soil bioturbation model[11] with associated soil mechanical properties and general biophysical traits of earthworms. Soil and climatic information are used to predict the global distribution of habitats and associated temporal windows of bioturbation activity. Although soil moisture and soil type dominate earthworm burrowing potential, other factors such as temperature[21], soil pH[22], and high sand contents[23] were considered.

**Earthworm bioturbation—cavity expansion model and soil mechanical properties**. Contrary to popular view, the primary mechanism for soil bioturbation by burrowing earthworms relies on their ability to penetrate and deform the wet soil matrix using their flexible hydroskeleton rather than ingesting POM-rich soil[13]. A recent biophysical model quantifies earthworm soil penetration and cavity expansion pressures[11]. The model defines the mechanical stress required for radial cavity expansion in an elasto-viscoplastic soil[11] that is linked with radial stresses $\sigma_r$ induced by the earthworm hydroskeleton at the cavity wall (Fig. 1). The minimal stress for cavity expansion in soil is given as

$$\sigma_r\left(R_p\right) = P_L - 2s_u \ln\left(\frac{R_p}{r_c}\right) = s_u \qquad (1)$$

where $r_c$ is the radius of the cavity, $P_L$ is the pressure at the cavity interface, $R_p$ is the radius of the elasto-viscoplastic interface (far-field), and $s_u$ is the soil shear strength. Solving for the cavity expansion pressure yields the following limiting pressure for soil deformation

$$P_L = s_u\left(1 + 2\ln\left(\frac{R_p}{r_c}\right)\right) = s_u\left(1 + \ln\left(\frac{G}{s_u}\right)\right) \qquad (2)$$

where $G$ is the shear modulus of rigidity. The ratio between the cavity zone and the viscoplastic zone converges to the ratio between the shear modulus and shear soil strength $\left(\left(\frac{R_p}{r_c}\right)^2 \to \left(\frac{G}{s_u}\right)\right)$ as the initial cavity radius approaches zero (e.g., when initiating the creation of a new burrow). Soil mechanical properties and soil moisture affect the model parameter values and thus the conditions that permit bioturbation by earthworms. We adopt a macroscopic rheological description of soil deformation[24,25] and use simplified power-law relations for linking soil mechanical properties to soil texture and water content similar to the work of Gerard[26] (Supplementary Information, Supplementary Figs. 1 and 2). The resulting expressions describe the minimum pressure an earthworm must exert to radially expand a cavity in soil (Fig. 1). Observations suggest that the earthworm hydroskeleton[27] can apply a maximum pressure of $P_w = 200$ kPa[28,29] (see Supplementary Information, Supplementary Fig. 3 for details regarding the sensitivity of $P_w$). In other words, earthworm bioturbation becomes mechanically impeded by soil mechanical conditions when $P_L(\theta, n) \geq P_w$, where $\theta$ is the soil water content, and $n$ is the summed fraction of silt and clay.

## Results

**Predicted earthworm hospitable regions**. We calculated mean annual cavity expansion limit pressures globally ($0.1° \times 0.1°$, monthly for 1981–2019) using the ERA5-land soil moisture reanalysis and SoilGrids[30] topsoil textural information (Fig. 2a). Different averaging methods were compared (Supplementary Fig. 4) and the harmonic average annual pressures are reported (Fig. 2 a). Geographical regions indicated in green are, on average, below the earthworm's biomechanical pressure limits. Independent data from a recent study[20] indicated less than 10% of observed earthworm abundance above a limiting pressure of 200 kPa (Supplementary Fig. 5). Additional factors that might exclude earthworm activity were considered to further constrain the predictions of potential earthworm habitats (Fig. 2b). Regions with subzero[21] mean annual temperature (MAT) are marked in

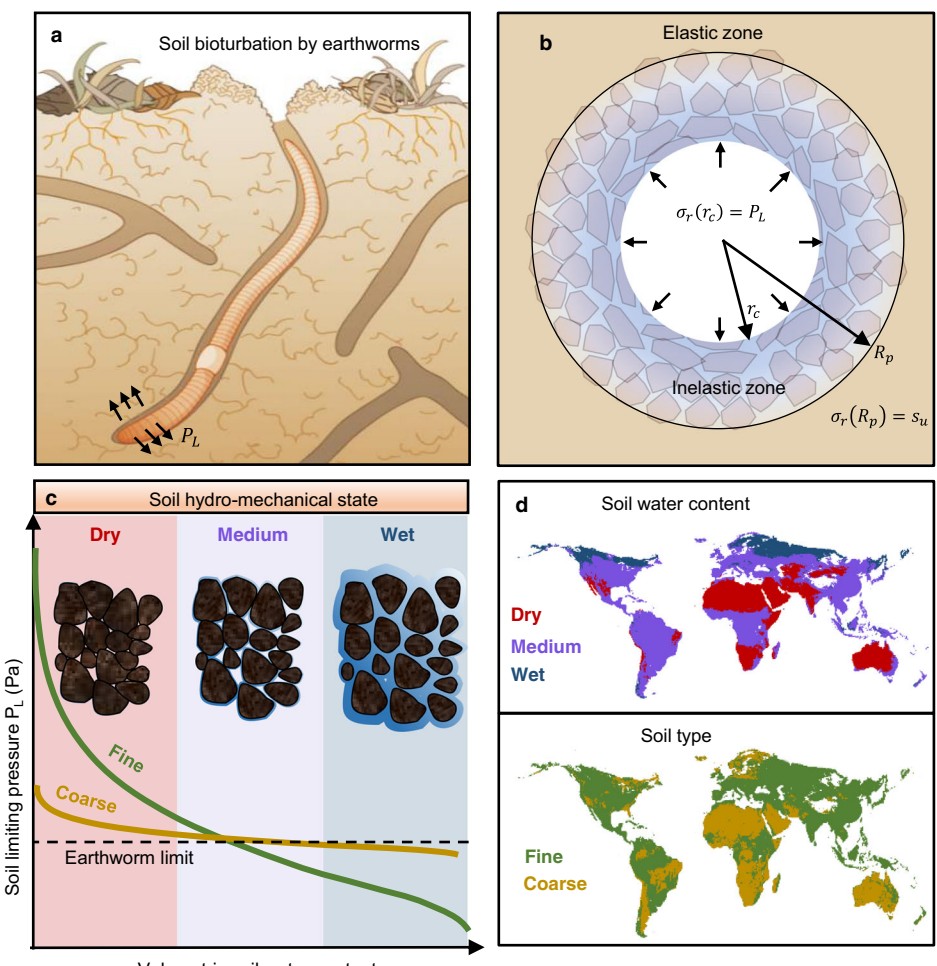

**Fig. 1 Earthworm bioturbation activity in structured soil. a** Subterranean bioturbation relies on earthworms' ability to mechanically penetrate and deform the soil using their flexible hydroskeleton, which is **b** modeled considering penetration and cavity expansion transverse to the earthworm body where radial stresses $\sigma_r$ exerted by the earthworm from the local cavity of size $r_c$. Yielding soil material is bounded by a remote elastic zone at a distance $R_P$ from the center of the cavity is dependent on **c** soil hydromechanical conditions that enable their hydroskeleton to form cavities. **d** Soil hydromechanical soil states can be mapped globally depending on soil water content and soil type, enabling inferences to earthworm distributions.

cyan. Regions where the soil pH is below 4.5[22] are indicated in magenta and regions where the soil sand content exceeds 80%[23] in yellow. For regions with pronounced seasonality, earthworms have developed ecological strategies to cope with periods during which soil mechanical conditions impede bioturbation (e.g., an extended period of dormancy[18,31]). Considering the minimal time window for a reproductive cycle and survival of newly hatched earthworms (total 4–6 weeks)[31], we required two consecutive months of favorable, soil mechanical conditions for permissible habitation. This would ensure at least one reproductive cycle per year[31]. Regions with shorter time windows are shown in orange (Fig. 2b). We note that these time windows may be sensitive to temperature (e.g., lower temperatures may require longer windows of activity). However, an in-depth analysis of this is outside of the scope of this study. Distributions of additional factors were compared to sites with earthworm occurrence from a recent study[14] (Supplementary Fig. 6). Comparing reported soil pH with values obtained from digital soil maps (SoilGrids[30]) revealed a narrower range of values than observed at the sample scale. Most occurrences of earthworms were reported for soil pH above 3.5 that mapped to SoilGrids[30] pH values above 4.5 (used for spatial mapping). Most sites with earthworm occurrence also received more than the previously reported[15] minimum mean annual precipitation (MAP) of 400 mm yr⁻¹.

We investigated the proportion of areas where masks based on auxiliary constraints overlap with those obtained from the cavity expansion limiting pressure $P_L$ using Jaccard indices (Fig. 2c). We can see that $P_{200}$ ($P_L < 200$ kPa) overlaps with 60% of the regions limited by MAT, 90% with activity, 60% with Soil pH, and 70% with sand content. Furthermore, we highlight the latitudinal regions where the respective constraints are most influential (Fig. 2d).

We conducted a sensitivity analysis which showed that considering both $P_{200}$ and activity did not significantly expand the inhospitable regions. Despite only a few regions where these auxiliary constraints uniquely limit earthworm activity (e.g., soil pH in the Amazon and MAT at northern latitudes), they were included to convey a more complete description of earthworm habitats. However, a few mechanisms based on the additional constraints remain unresolved (e.g., low soil pH might also be a proxy for frequently flooded soils with reduced oxygenation).

**Modeled and observed earthworm global distributions.** Detailed comparison of regions with ample observations was used for model evaluation. For example, earthworm spatial distributions for Australia and North America are depicted in Fig. 3a, b, respectively[32]. The large extent of arid regions in Australia limits

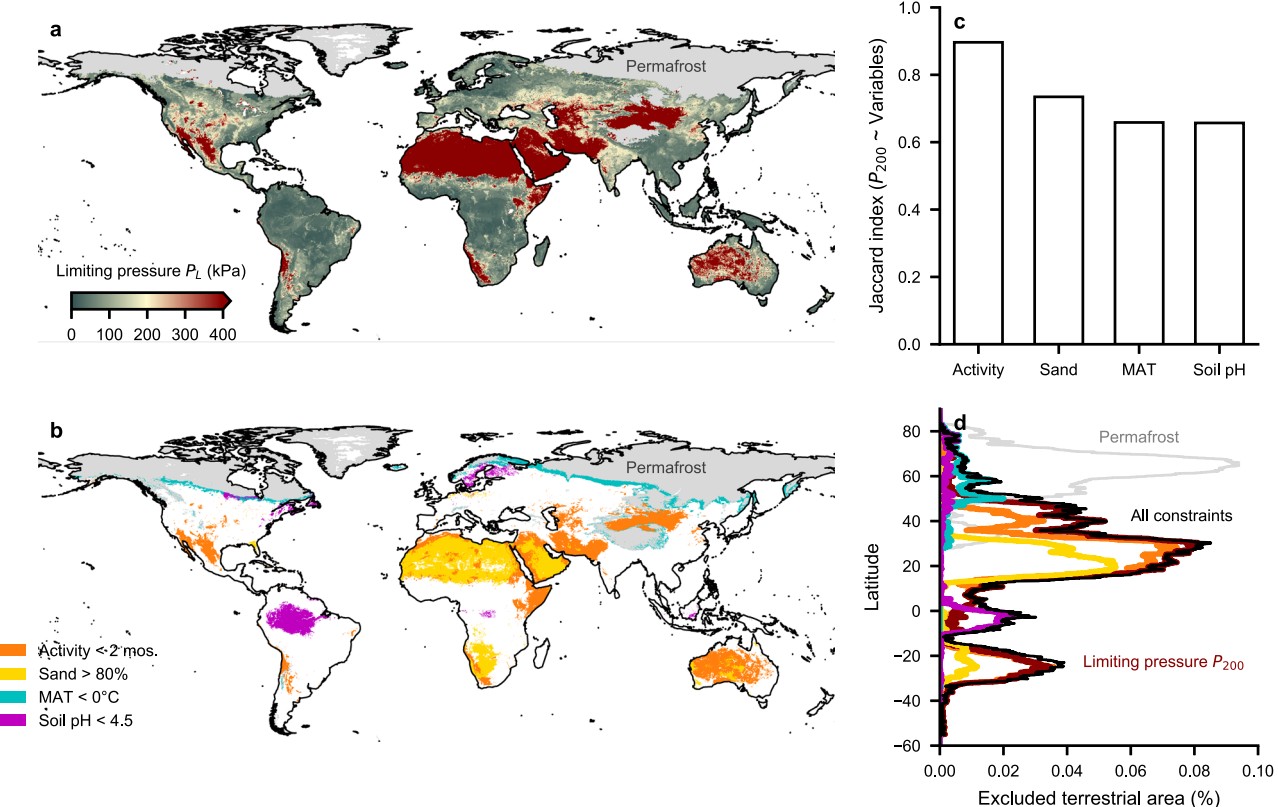

**Fig. 2 Global map of earthworm hospitable zones. a** Green regions indicate that annual average pressures required for cavity expansion are below the earthworm's hydrostatic pressure limit (200 kPa). Pressures are truncated to values below 400 kPa for visualization (dark red) and permafrost regions were removed (gray). **b** Other factors that may impede earthworm activity. Cyan regions indicate the sub-zero mean annual temperature (MAT), magenta regions mark soil pH < 4.5, yellow regions indicate coarse soil texture (sand content > 80%), and orange regions indicate that there are fewer than two consecutive months during which the soil mechanical properties permit cavity expansion. Regions of different limiting factors may overlap and were ordered for visibility. **c** Overlap in the area (Jaccard index) that is considered hospitable based on pressure below 200 kPa ($P_{200}$) compared with other variables. **d** Latitudinal distribution of terrestrial area that is excluded by considering each variable independently (colored lines) and the fully constrained habitat area (black).

earthworm activity to the coasts that receive sufficient rainfall to moisten the soil. This is in good agreement with model predictions as shown with the 400 mm yr$^{-1}$ contour of MAP[15] (Fig. 3a). For North America, the model predicts that earthworm activity is possible from the east coast to the Midwest followed by a sharp decrease in occurrence until the west coast (Fig. 3b). These trends are similar to previously estimated earthworm distributions[16] with a sharp cutoff near arid regions. Around half of the terrestrial surface (>−60°N) permits earthworm activity but most observations of earthworm presence originate from Europe (Fig. 3c). Reported earthworm presence agreed with model classification for 86% of the geographical occurrences (global within 0.1° × 0.1°, $n = 7346$). Although there were 13% of false negatives, these were often associated with local geographical features (e.g., river banks and anomalous precipitation zones) as depicted in Fig. 3. To test the robustness of classification and its sensitivity (hit-rate) we performed random re-sampling of occurrences with replacement (Supplementary Fig. 7).

**Earthworm seasonal activity windows**. The global map of average conditions conducive to earthworm burrowing activity conceals the nuanced dynamics associated with seasonal activity windows that are driven primarily by precipitation[19,33]. To provide a succinct picture of this ingredient, temporal activity windows (seasonality or wet periods) for earthworms are illustrated in Fig. 4. The temporal variability of limiting soil pressures is described spatially by the coefficient of variation and highlights

regions in which the impact of seasonality on earthworm activity is most pronounced (Fig. 4a). Figure 4b presents the median limiting pressure across latitudes for a climatic year (i.e., an ensemble year considering several decades). This highlights the dynamic nature of soil hydromechanical conditions that constrain seasonal earthworm activity and delineates regions where soil conditions prohibit earthworm activity year-round (i.e., arid regions).

To evaluate the temporal resolution of our model predictions, we used monthly earthworm abundance data spanning from 2002 to 2008[34] for comparison with modeled dynamics of cavity expansion limiting pressures (Fig. 4c). What can be seen between the two curves is that peaks in $P_L$ correspond with troughs in abundance. These contrasts appear to correspond well at a monthly resolution and suggest that our model can resolve seasonal dynamics. Furthermore, at two peaks in $P_L$ that come close to the 200 kPa threshold (2003–2004 and 2006–2007), we see that earthworm abundance approaches zero.

The required minimal cavity expansion pressures are compared for two contrasting biomes where MAT, sand content, and pH, were not limiting. A grassland located at 9.55°N, 14.65°E and a desert located at −22.95°N, 132.95°E are indicated in Fig. 4 a. Results suggest that soil moisture content mediated by precipitation facilitates mechanical activity for as much as 4.5 consecutive months in the grassland (Fig. 4d) while the infrequent precipitation in the desert (Fig. 4e) resulted in no appreciable temporal activity window for bioturbation or reproduction.

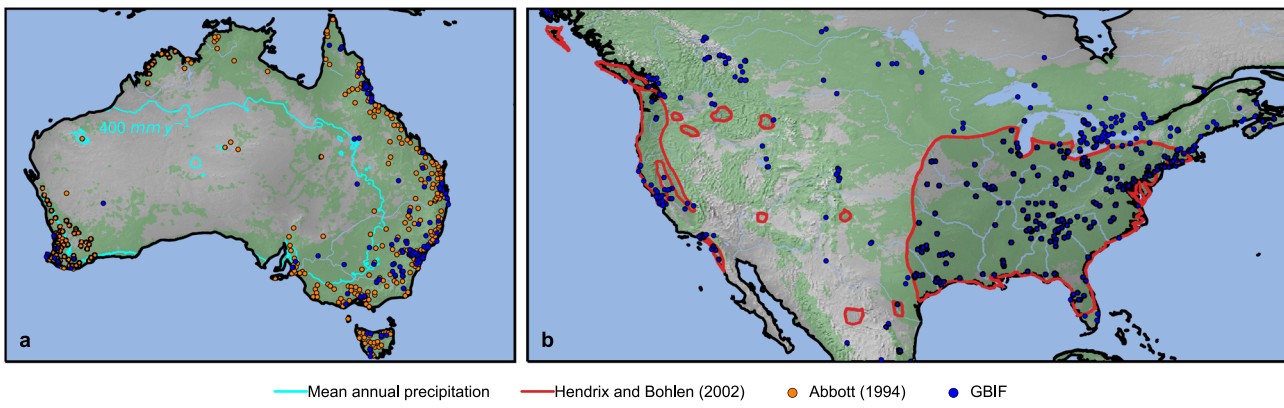

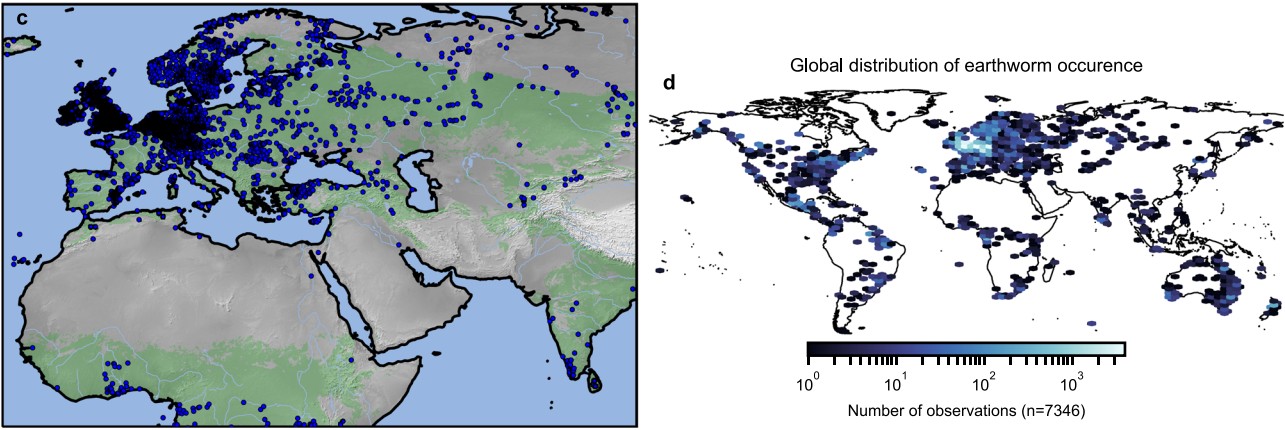

**Fig. 3 Comparison of predicted hospitable zones and reported earthworm distribution. a** Potential earthworm habitats (green) including soil hydromechanical limitations for Australia. Locations with reported presence of earthworms from two datasets are displayed; GBIF (blue points) and Abbott[15] (orange points). Regional limitation of earthworm activity is delineated by 400 mm yr$^{-1}$ of mean annual precipitation[48] (cyan contour) as previously reported[15]. **b** Predicted earthworm habitats for North America. Observed occurrences (Global Biodiversity Information Facility, GBIF) are in good agreement with regional extents of earthworm communities (redrawn from Hendrix and Bohlen[16], red). **c** Regions in East Eurasia and Northern Africa that could support earthworm soil bioturbation. **d** Global distribution of earthworm occurrence. Made with Natural Earth. Free vector and raster map data @ naturalearthdata.com.

Lastly, we compared species richness reported in Phillips et al.[14] to the fragmentation of habitats across latitudes (Fig. 4f). Latitudinal habitat fragmentation was measured by counting the number of land fragments that are broken up by inhospitable zones and water bodies within a 0.1° wide strip around the globe. Results suggest higher species richness with an increased number of fragmented habitats at the spatial resolution of ~10 km.

## Discussion
A novel biomechanical model (to the best of our knowledge) for earthworm bioturbation in combination with climatic and soil conditions enabled mapping of global habitat suitability (Fig. 2) and comparison with earthworm distributions (Fig. 3). Favorable soil moisture and mechanical conditions dominate the global distribution of earthworms. Additional constraints such as permafrost soil and subzero MAT[21] preclude earthworm activity in large parts of the world. Despite evidence for soil acidity limitations (soil pH < 4.5)[22], the global earthworm distribution was not overly sensitive to low values of soil pH[16]. The primary mechanism[14] that shapes earthworm occurrence appears to be driven by soil physical (hydromechanical) conditions; determined by soil moisture and earthworm physiological limitations in unfrozen soils.

The distributions of environmental conditions associated with earthworm occurrence compare favorably with the range of values reported in a recent global study[14] (Supplementary Fig. 6). The modeled soil limit-pressures appeared to also correspond strongly with observed earthworm abundance using independent data (Supplementary Fig. 5). However, modeled trends at ~10 km resolution preclude representation of many small-scale niches. For example, river corridors that cut across arid regions in the US Midwest reported the presence of earthworms not represented by the model. Other examples were found along rivers in South-East Australia and Eurasia. Similarly, inhospitable regions with low soil pH and high sand content may not be properly captured by the smoothed estimates of digital soil maps[30] as evident when comparing with values reported for soil samples (Supplementary Fig. 6a, b). We note that many biological and chemical soil properties are also related to climatic hydration conditions[31,35]. Our results represent average climatic tendencies manifested across biomes and spatial scales (~10 km resolution). Such global estimates might average out locally limiting factors (soil moisture, soil compaction, temperature, and soil pH), thus contributing to model predicted false negatives. Furthermore, our estimation for maximal earthworm hydroskeletal pressures is based on earthworms residing in temperate regions[28]. Large earthworms found in the tropics or Australia may exert greater pressures and could

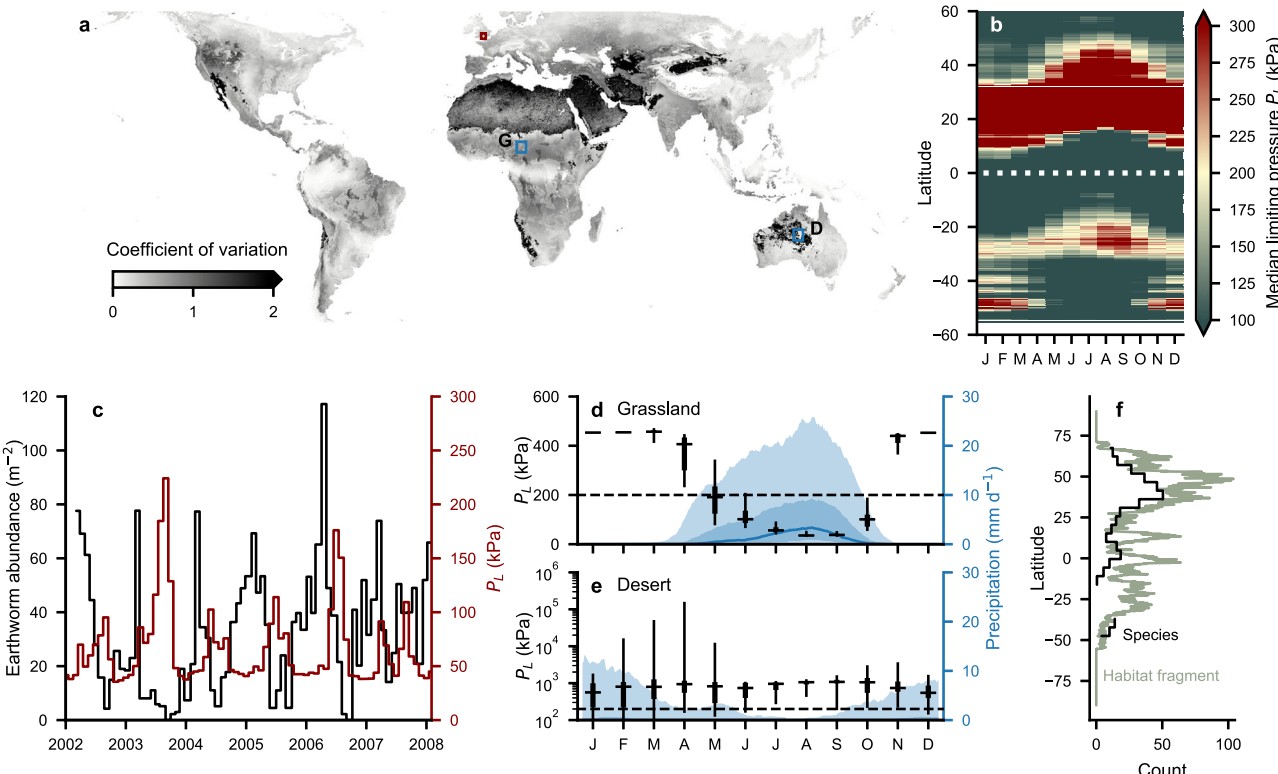

**Fig. 4 Temporal windows of potential earthworm burrowing activity. a** Global map of temporal hydromechanical variations (coefficient of variation of limiting pressures). **b** Median earthworm limit pressures across latitudes for a climatic year. **c** Time series comparison of modeled cavity expansion limiting pressure (red) with measured earthworm abundance (black). Earthworm abundance was measured monthly in the New Forest, Hampshire UK (5.9°N, −1.6°E,[34]) over six consecutive years. **d, e** Median climatic limiting pressures (boxes indicate central 50 and 90% of values) required to burrow through the soil are associated with mean daily precipitation[48] (blue line and shading; 30 days running median and central 50 and 90% of values) for (**d**), a grassland (**g**: 9.55°N, 14.65°E) and (**e**), a desert (**d**: −22.95°N, 132.95°E) as indicated in (**a**). The maximum radial earthworm pressures $P_w$ (dashed line) are shown. Soil limit pressures are reported for the topsoil (0–7 cm) and are assumed to represent the driest part of the soil profile. **f** Habitat fragmentation based on habitable regions is plotted in comparison with species richness[14] results for different latitudes. The maximum radial earthworm pressures $P_w$ (dashed line) are shown. Soil limit pressures are reported for the topsoil (0–7 cm) and are assumed to represent the driest part of the soil profile.

thus be less limited. We note that there are several challenges when trying to relate earthworms' hydroskeletal pressures with earthworms' body mass. Pressures translate to stress. Although pressures are often confounded with forces, these do not necessarily scale similarly with earthworm size. Earthworms with larger biomass are likely to exert greater forces, yet, this may not necessarily translate to higher pressures. Large anecic earthworms (*L. terrestris*) consistently demonstrated to exert lower pressures compared with smaller endogeic earthworms across several studies[28,29,36]. However, we are not precluding the possibility of earthworms that may exert higher pressures (e.g., large worms found in Australia). As more information becomes available, the spatial extent and constraints can be easily revised based on the mechanistic model. Furthermore, given a model sensitivity analysis on the limiting pressure (Supplementary Fig. 3), we are confident in our current model predictions.

Moreover, it remains challenging to address potential observational bias in the spatial patterns of reported earthworm occurrences. Most occurrences are reported for few countries in Europe (United Kingdom, Germany) resulting in strong spatial clustering of presence data that hampers the assessment of model sensitivity (hit-rate). By considering the observation density and performing weighted, random resampling we observe a minor reduction in hit-rate (from 86 to 84%) and find that average estimates are robust against variations in sample size (Supplementary Fig. 7). While this may not fully resolve the issue of observational bias, we can analyze possible tendencies of reduced

sensitivity. Overall, the lowest hit rate is still well above 50%, which would be expected by a coin toss and, coincidentally, by the fraction of terrestrial area that is predicted to be hospitable to earthworms.

In addition, the seasonality of limiting soil pressures defines temporal windows of earthworm activity and selects for particular ecological life strategies. The model predicted activity windows (Fig. 4) correspond closely to previously reported seasonal variations in earthworm communities[17,18]. This suggests that their ecological strategies (i.e., dormancy cycles and reproduction cycles) are mediated by soil hydromechanical factors. While the shortest possible temporal window that supports thriving earthworm communities is unknown, a sufficiently long window is required for earthworm annual reproduction[18]. Earthworms may live several years, but the fertilization and egg incubation takes 3–4 weeks[18,31]. In addition, young earthworms need a few weeks to build up biomass to survive dormancy[18,31]. We could assume 1–2 months of favorable conditions to be the minimum requirement for survival and reproduction[31]. Narrow windows would also limit earthworms' accessibility to plant-derived POM, which could further preclude their activity in deserts with low net primary productivity (Fig. 4c, d). Strong seasonal variation poses further constraints on earthworm activities linked to the variability of limit pressure (Fig. 4a). Although we present harmonic averaging that provides more inclusive bounds for earthworm habitats in regions with strong seasonal variation (e.g., Spain, Fig. 3; for comparison of averaging methods see Supplementary

Fig. 4), the mechanistic model allows for quantification of the seasonal variability in earthworm habitats (Fig. 4 a). Despite few regions of high volatility, climatic predictions are robust for most regions. For example, permissive regions of earthworm activity in Asian islands such as the Philippines[37] are predicted.

Furthermore, our results quantify the dynamics of latitudinal patterns (Fig. 4b). While particular regions remain stable (i.e., favorable or uninhabitable), several latitudes exhibit strong fluctuations. One of the more striking features is observed between 20°N and 30°N. These zones are characterized by particularly harsh conditions. Interestingly, the highest number of earthworm species was reported for this range[14]. Compatibility between the two results would suggest that species richness is high under environmentally harsh conditions (Fig. 4e). However, taking the latitudinal median might miss small regions that permit earthworm burrowing activity. The limited spatial extent of such "patches" would not allow for widespread migration and favor endemic (isolated) populations; resulting in high species richness over climatic timescales. Nonetheless, this is not to suggest that the short-term anthropogenic fragmentation of earthworm habitats would promote species diversity.

The study provides a framework for the prognosis of potential migration trends, climatic barriers, and the promotion of sustainable land use. Regions of North America with limited earthworm activity are predicted by our model in agreement with previously reported earthworm distributions (Fig. 3). Isolation of earthworm communities in North America could be attributed to drier regions central-westward that act as geographic barriers. These regions obstruct earthworm migration and could explain why few native earthworm species returned to North America post glaciation[14].

The growing threat of soil compaction associated with increased land use intensification[38] is motivating a large push towards no-tillage practices[9,38]. Regions that indicate soil bioturbation potential by earthworms may be used to further prompt more sustainable agricultural practices[39], which would reduce the frequency and intensity of tillage machinery while maintaining soil structure suitable for crop growth[38]. The modeled regions of bioturbation potential are based on first principles that are independent of earthworm occurrence or abundance data and can serve as a reference for evaluating agricultural practices across biomes.

While the mechanistic approach presented in this study requires a more nuanced understanding of the underlying principles that facilitate earthworm occurrence, our methodology provides several advantages to correlative techniques (See Supplementary Information and Supplementary Fig. 8). Our model can highlight where correlative models are violating causal processes and disentangle constraint collinearities, which is not feasible with correlative modeling[40]. As a result, our model circumvents excessive speculation that can lead to incomplete or invalid inferences (e.g., understating the importance of soil physical properties)[14]. The modeling framework (Fig. 3) could be readily incorporated in climate models with minor computational costs to represent dynamics of global earthworm habitats and activity windows[41]. Unlike a static picture of global distributions[14,42], the model could be used to assess future trends in regions viable for agriculture and land use management (tillage vs. no-tillage) concerning earthworms' contributions to soil structure. Predictions of earthworm activity and migration patterns could be linked to the future expansion of wetter (or drier) regions. The results presented in this study remain tentative awaiting additional direct observations (beyond "presence-absence"). For example, we envision experiments across soil moisture gradients, or soil strength (i.e., compaction) for similar soil types and plant cover to test the role of mechanical constraints under natural and prescribed conditions. Although the focus has been on hospitable regions for earthworm activity, soil water contents associated with limiting earthworm pressures have been shown to affect plant root growth for many soil types. Bengough et al.[43] reported that this lower bound in soil moisture provides favorable mechanical conditions and water availability for plant roots. This becomes evident when considering global gross primary production (GPP), which highlights very similar spatial patterns[44] compared to predicted earthworm habitats. Furthermore, plant roots could benefit from mutualistic interactions with earthworms[5], thus finding benefits from regions where earthworms thrive and vice versa.

Although comparisons made in this study inspire confidence in our model, refinements would be needed to better predict bioturbation and foraging activity. We envision, development of population densities based on energetic considerations that include soil carbon input fluxes[35] (e.g., GPP). Reported earthworm populations range between 60 and 350 individuals per m$^2$ of soil surface[45] and, likely, resource availability (i.e., soil organic carbon or POM) could limit earthworm abundance in particular regions. Considering such factors in a mechanistic modeling framework would help disentangle the various effects of organic matter accumulation on soil mechanical properties (bulk density), soil water characteristics (water retention), and physiological (energetic) constraints. Such refinements would enable the model to generate estimates regarding earthworm abundance, which is beyond the scope of the current study.

Insights into the fundamental principles that shape earthworm ecological trends as reported in previous studies[14–16] place such empirical observations on a mechanistic basis. This deepens our understanding of the processes relevant to predators, soil flora and microbes that interact with earthworms, and the general ecosystem services that earthworms provide[5]; all of which are built on the foundations of soil hydromechanical status.

## Methods

**Earthworm limiting pressure and activity windows**. Using global soil moisture data combined with the critical soil hydromechanical states that limit earthworm burrowing, we determined climatic regions that could support potential earthworm bioturbation activity. Regions with a high likelihood of permafrost are removed from calculations (with permafrost zonation index[46] exceeding 0.1). For each geographic location, we evaluated the parametrized model using soil textural information from SoilGrids digital soil maps[30] and monthly averaged soil moisture estimates from ERA5-land (https://doi.org/10.24381/cds.68d2bb30)[47]. All global raster data were harmonized to a common grid of 0.1° resolution (~10 km) using nearest-neighbor interpolation of the uppermost soil depth layer (0–5 cm and 0–7 cm for SoilGrids and ERA-5 land, respectively). The limiting pressure (Eq. (2)) was calculated for the entire record of the ERA5-land dataset that ranges from 1981 to 2019 at a monthly resolution. Based on the limiting pressure time series, we estimated the number of consecutive months below 200 kPa and the ensemble average pressure for every grid cell. A comparison of averaging methods is reported in the Supplementary Information, and we reported harmonic averages throughout the main text. A monthly resolved time series record of earthworm abundance in the New Forest, Hampshire UK (5.9°N, −1.6°E) spanning from 2002 to 2008 was taken to illustrate the dynamic resolution of our model predictions[34]. Separately, two specific regions were selected to illustrate temporal activity windows within a year for a given biome: a grassland located at 9.55°N, 14.65°E, and a desert located at −22.95°N, 132.95°E. We aggregated the limiting pressure time series to climatic monthly values and compared it with daily climatic precipitation estimates obtained from MSWEP[48]. Daily precipitation estimates were smoothened using a 30-day rolling average for comparison with monthly pressure values and to delineate time windows of earthworm burrowing activity.

**Additional factors that impede earthworm activity**. Climatic factors and soil properties were used to illustrate additional factors that could impede bioturbation activity by defining thresholds for earthworms' tolerance. Regions, where the MAT was below zero, were considered zones of impedance. Besides the soil mechanical impedance becoming augmented in a manner not currently considered in our model, these low temperatures were considered to decelerate earthworms' metabolic cycles to critical states[21], which may ultimately lead to earthworms freezing. Besides soil temperature, low soil pH has often been cited as being critical for earthworm habitat suitability[14]. We outlined global regions where soil pH is below 4.5[22,31]. Regions, where sand content exceeded 80%, were also considered as

regions of impedance. Although there are sandy soils where earthworms have been observed (e.g., dunes in the UK[49]), the abrasive nature of sand grains is typically obstructive[50]. We note that soil organic carbon and POM would also play a role in limiting earthworm abundance. However, as they are likely to co-occur in hydromechanically hospitable conditions, we focused our study on physical and chemical factors impeding potential earthworm activity.

**Earthworm occurrence data**. We compared our theoretically determined regions with previously published empirical maps that outline earthworm distributions for Australia[15] and North America[16] and with presence-only data of ten earthworm species (*Almidae, Eudrilidae, Glossoscolecidae, Hormogastridae, Lumbricidae, Microchaetidae, Moniligastridae, Ocnerodrilidae, Octochaetidae, Sparganophilidae*) as deposited in the Global Biodiversity Information Facility (GBIF) database (https://doi.org/10.15468/dl.xstqow [51], https://doi.org/10.15468/dl.wghggg[52], https://doi.org/10.15468/dl.3yj8pk[53], https://doi.org/10.15468/dl.lzuwlg[54], https://doi.org/10.15468/dl.vwqtsk[55], https://doi.org/10.15468/dl.brqmht[56], https://doi.org/10.15468/dl.ghccto[57], https://doi.org/10.15468/dl.dk97gk[58], https://doi.org/10.15468/dl.xjw6kc[59], https://doi.org/10.15468/dl.9a4ojx[60]). The distribution of each species occurrence was shown in Supplementary Fig. 9.

**Reporting summary**. Further information on research design is available in the Nature Research Reporting Summary linked to this article.

## Data availability

All data used in this study is available from public sources. The generated global time series of soil limiting pressure is deposited in a public repository https://doi.org/10.3929/ethz-b-000476615[61]. Additional source data of climatic limiting pressures for Fig. 4 d-f is provided as Supplementary data. Presence-only data of ten earthworm species (*Almidae, Eudrilidae, Glossoscolecidae, Hormogastridae, Lumbricidae, Microchaetidae, Moniligastridae, Ocnerodrilidae, Octochaetidae, Sparganophilidae*) can be found in the Global Biodiversity Information Facility (GBIF) database (https://doi.org/10.15468/dl.xstqow[51], https://doi.org/10.15468/dl.wghggg[52], https://doi.org/10.15468/dl.3yj8pk[53], https://doi.org/10.15468/dl.lzuwlg[54], https://doi.org/10.15468/dl.vwqtsk[55], https://doi.org/10.15468/dl.brqmht[56], https://doi.org/10.15468/dl.ghccto[57], https://doi.org/10.15468/dl.dk97gk[58], https://doi.org/10.15468/dl.xjw6kc[59], https://doi.org/10.15468/dl.9a4ojx[60]).

## Code availability

Code pertaining to the soil mechanical mapping from the soil texture and moisture status are provided in https://doi.org/10.3929/ethz-b-000476615[61] under the name pressure_timeseries.py. The script was developed in Python 2.7.15.

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

## Acknowledgements
This research was carried out at ETH Zürich and the University of Southampton. The authors would like to acknowledge the help from Dr. Peter Lehmann for preliminary soil moisture maps, which were crucial to motivating this study. The authors acknowledge helpful discussions with Prof. Ning Lu regarding soil mechanical properties and thank Dr. Katherine Williams for proofreading the document.

## Author contributions
S.R. and D.O. conceptualized the study. S.R. developed a soil bioturbation mechanical model. S.R. collected preliminary data pertaining to earthworm activity windows. S.R. generated an initial paper draft. S.R. and S.B. developed global distribution maps. S.B. and D.O. included auxiliary constraints. S.B. developed code to generate robust maps from available soil data. S.B. provided a depth comparison between mechanistic and data-driven modeling approaches.

## Competing interests
The authors declare no competing interests.
