## [Peer Review File · Communications Biology]

Reviewers' Comments:

Reviewer #1:

Remarks to the Author:

I really enjoyed reading this paper. It applies a mechanistic model of the ability of earthworms to move through soil to develop a prediction of habitat suitability of earthworms across the globe. It is of broad significance given the important positive effects earthworms have on the agricultural productivity of soils. I think the paper is written well and explains the underlying principles being used. Overall the physical arguments for hydromechanical limits on earthworms are well made and the model is well constructed.

Where I think the paper is weak is in how it assesses model performance and parameter choices, as well as some of the additional constraints applied to predict distribution. There are generally two paths that can be taken with modelling habitat suitability, a correlative approach and a mechanistic approach. The authors are going down the mechanistic path which has the advantage of being able to make robust predictions across a wide range of environments without extrapolation issues.

However, the predictions of such a model need to be based on good arguments for parameter values so that it can be strongly inferred where conditions would be physically out of bounds. Ideally, the authors would provide data of actual burrowing activity in earthworms as a function of measured soil conditions to show that their choices of parameters, especially the P_w threshold, really do capture hard physical constraints on the earthworms, and second to see how at the same site the gridded data they used produce a similar prediction.

Instead the authors relate their predictions to observed occurrences, which is a much weaker test and I'm not left convinced that what they have done is more powerful than a more correlative approach, especially given the various additional constraints that they have added on. How much are the predictions improved as you add the various additional constraints beyond the pressure limit? Note that the activity limit should really be made to be a function of soil temperature as the speed of the earthworm life cycles will be very temperature-dependent.

I agree that the authors have generated a powerful index for inferring earthworm limits but they could potentially make the mechanistic aspect of their work stronger if they had a more convincing validation. The other way to approach this, especially if there are no time-series available of earthworm activity to test the model with, is to make predictions against some non-mechanistic benchmarks, e.g. that don't account for soil properties but only soil moisture. The main goal should be to more convincingly show why this approach is better than a more statistical approach, especially in regards to inferring what will happen under novel conditions.

The authors could also potentially compare their predictions against date-specific observations given that they are using historical forcing data. And related to this, with regard to the monthly averaging, why not just compute the hourly movement potential and summarise those in some way (e.g. maximum runs of movement or maximum runs of no movement) rather than applying the averaging methods? This could at least be done for a subset of site to assess the effect of the monthly averaging.

I found the discussion to be a bit long and extending beyond the analyses too much. The discussion could include more about the advantages and disadvantages of the approach taken over more correlative approaches, and cut down on the general importance of earthworms for soils (which is made in the intro).

Finally, the authors might find the following articles of interest in the context of mechanistic modelling and microclimates (there's quite a lot of other papers on this general issue of correlative and mechanistic modelling):

Buckley, L.B., Urban, M.C., Angilletta, M.J., Crozier, L.G., Rissler, L.J. & Sears, M.W. (2010) Can mechanism inform species' distribution models? *Ecology Letters* 13, 1041–1054.

Dormann, C.F., Schymanski, S.J., Cabral, J., Chuine, I., Graham, C., Hartig, F., Kearney, M.R., Morin, X., Romermann, C., Schroder, B. & Singer, A. (2012) Correlation and process in species distribution models: bridging a dichotomy. *Journal of Biogeography* 39, 2119–2131.

Kearney, M.R. & Porter, W.P. (2009) Mechanistic niche modelling: combining physiological and spatial data to predict species' ranges. *Ecology Letters* 12, 334–350.

Kearney, M.R. (2019) microclimUS: hourly estimates of historical microclimates for the United States of America with example applications. *Ecology* 100, e02829.

Kearney, M.R., Gillingham, P.K., Bramer, I., Duffy, J.P. & Maclean, I.M.D. (2020) A method for computing hourly, historical, terrain-corrected microclimate anywhere on earth. *Methods in Ecology and Evolution* 11, 38–43.

Abstract

line 17 - change 'promotes' to 'alters' or 'affects', or be specific about the context in which it 'promotes' functioning (e.g. agricultural)

line 26 - change 'migration' to 'range expansion' or something similar unless you mean potential to get to new places rather than to persist once arrived

Main

line 29 - change 'sustains' to 'alters' or 'influences' or something similar

line 32 - change to 'improve groundwater recharge and soil water retention, and'

line 37 - add 'of' before 'up to'

line 43 - earthworms' (i.e., add possessive apostrophe)

line 71 - put the '(iii)' after 'to'

line 99 - it would be good if more information can be provided on this figure as it is such an important threshold - how sensitive are the results to variation in it?

Results

- line 117 - this time window will be strongly temperature sensitive

- line 152 - is a 'climatic year' any different to a 'year'?

Methods

- generally check the tense used throughout the methods, including the supplement, and keep it past tense, e.g. the first sentence is past tense and then the next two are present tense

line 291 - change 'upper most' to 'uppermost'

line 309 - "decelerate earthworms' metabolic cycles to critical states" is vague and the argument is not strong - I think the low temperature constraint needs some further justification/exploration

Supplement

- line 35 - remove second period after 'rate'

- line 43 - Make this a full sentence or remove the capital L in 'leading'

- line 45 - change "earthworms hydroskeleton" to "an earthworms' hydroskeleton"

- line 56 - change 'condition' to 'conditions' and add 'by' after 'or'

- line 70 - "The relations of coefficient to soil fine fraction" is not a full sentence

- line 78 - 'fit' should be 'fitted'

Reviewer #2:

Remarks to the Author:

The paper presents a model of the physical and other environmental limitations to earthworm

activity, the former being novel and allowing new insight into earthworm ecology. The model is implemented to predict earthworm distribution at a global scale and provides an important contribution to the field. The study is generally well conducted, well written and presented with some brilliant figures. The work is original and of wide interest to the field. However, the study does fall short of some of its larger claims. These are outlined below under 'major comments', followed by some more 'minor suggestions'.

Major comments:

1) 'Mechanistic model': I have some issues in the claim that the model is mechanistic. While the limiting pressure has been parameterised based on some first principles (Eq. 1 & 2) the model itself determines whether earthworms will be present or not given five environmental limits. The results of each environmental limit are also not entirely clear and so it is difficult to disentangle the major controls on earthworm distribution.

2) The limiting pressure has been inferred for all earthworms, regardless of their body size – which varies massively and will be directly associated with hydrological functioning. Surely there is error associated with this assumption, but it is not mentioned.

3) Following on from a comparison of environmental factors on earthworm distribution in comment 2, no meaningful metrics are given for the model's goodness of fit to the observed data. By providing such metrics, and a breakdown of the metric of each environmental factor in isolation this would be more useful in some ways than the maps.

4) The abstract claims the model's applicability to investigate how land use change alters earthworm distribution, but no analysis has been conducted to investigate this.

Minor suggestions:

L17: Earthworm

L17-L20: I would suggest linking the two sentences together for a concise introduction e.g. Earthworm activity modifies soil structure and promotes hydrological ecosystem functions, but this relationship is constrained by soil hydromechanical conditions.

L21: 'developed here'?

L21-22: Find this quite confusing – do you mean 'A novel biophysical model is developed here to link the biochemical limits of earthworm activity to predict bioturbation...'?

L22: Suggest replacing 'inject' with 'include' or 'model'.

L24-25: Any metric for this?

L25-26: Why? Major environmental controls are?

L26-27: Again, is there more information to give here? In particular regions or land use changes?

Extended figs 1 + 2: Is there any way of relating 'silt+clay (%)' to more commonly used bulk density? Or maybe there could be a sentence in the MS to make the link clear?

Figure 1: Brilliant figure! Minor points – the box around 2b is slightly thinner than the others, and 'Soil type' in 2d is cut off slightly.

Extended Fig 3 would benefit from larger text to make it readable.

Fig 2: L478 – purple rather than blue, and magenta rather than red? Also find the distinction between green and cyan a bit too similar and would suggest an alternative colour for one.

L112-114: Seems as though these values have been plucked out the air to make the modelled distributions fit the observations – please provide references or indicate that the evidence is provided elsewhere.

L147-148: 'seasonal activity windows that are driven primarily by precipitation' – I think others might disagree with the seasonal drivers so please provide a reference for this information or indicate how you inferred it.

Extended Fig 7 – do all data coordinates correspond to land surfaces? It seems that some data points are in the middle of the ocean.

Data availability – would expect the mapped predictions to be made openly available.

We have provided our point-by-point rebuttal to the reviewers' comments (presented in **bold, italicized, and underlined Times New Roman font**) and our response in in blue Calibri.

Referee expertise:

Referee #1: Biophysical ecology

Referee #2: Soil ecology, ecological modelling

Reviewers' comments:

Reviewer #1 (Remarks to the Author):

I really enjoyed reading this paper. It applies a mechanistic model of the ability of earthworms to move through soil to develop a prediction of habitat suitability of earthworms across the globe. It is of broad significance given the important positive effects earthworms have on the agricultural productivity of soils. I think the paper is written well and explains the underlying principles being used. Overall the physical arguments for hydromechanical limits on earthworms are well made and the model is well constructed.

Where I think the paper is weak is in how it assesses model performance and parameter choices, as well as some of the additional constraints applied to predict distribution. There are generally two paths that can be taken with modelling habitat suitability, a correlative approach and a mechanistic approach. The authors are going down the mechanistic path which has the advantage of being able to make robust predictions across a wide range of environments without extrapolation issues.

Thank you very much for the encouraging comments. We hope that the amendments made in the manuscript adequately address shortcomings of our model evaluation, parameter choices, and additional constraints.

01.1

However, the predictions of such a model need to be based on good arguments for parameter values so that it can be strongly inferred where conditions would be physically out of bounds. Ideally, the authors would provide data of actual burrowing activity in earthworms as a function of measured soil conditions to show that their choices of parameters, especially the P_w threshold, really do capture hard physical constraints on the earthworms, and second to see how at the same site the gridded data they used produce a similar prediction.

A1.1.

To place the selection of parameters in perspective, we rely on rigorous models and previously reported measurements of earthworm hydroskeletal pressure thresholds

from the literature (McKenzie and Dexter, 1988 , Quillin, 1999)^{1,2} and our own studies (Ruiz and Or, 2018)³. Based on these results that were generalized to¹⁻³ different soil types and conditions, we are reasonably confident that cavity expansion limiting pressures by earthworms and the soil and hydration specific parametrization reflect real and consistent physical constraints on earthworm (burrowing) activity. To the best of our knowledge this the first model to offer predictions based on first principles (and not statistical mapping of observations). Additionally, the spatial estimates of soil pressure limit are in good agreement with observed earthworm occurrence patterns of Abbott (1994)⁴ and Johnston (2019)⁵ as also shown in Extended Data Fig. 4.

We fully agree that direct comparisons with observations of actual burrowing activity would have been ideal, however, such information is presently unavailable (we hope that this and similar studies can guide future testing and verification studies). The best available information (other than direct measurements reported in Ruiz and Or, 2018)³ was included here, namely monthly time series of earthworm abundance for comparison with the dynamics of limiting soil pressures in Fig. 4c supported by the following explanatory text:

“To evaluate the temporal resolution of our model predictions, we used monthly earthworm abundance data spanning from 2002 to 2008 (Eggleton et al. 2009)⁶ for comparison with modeled dynamics of cavity expansion limiting pressures (Fig. 4c). What can be seen between the two curves is that peaks in P_L correspond with troughs in abundance. These contrasts appear to correspond well at a monthly resolution, and suggest that our model can resolve seasonal dynamics. Furthermore, at two peaks in P_L that come close to the 200 kPa threshold (2003-2004 and 2006-2007), we see that earthworm abundance approaches zero. “

Fig. 4: Temporal windows of potential earthworm burrowing activity. **a**, Global map of temporal hydromechanical variations (coefficient of variation of limiting pressures). **b**, Median earthworm limit pressures across latitudes for a climatic year. **c**,

Time series comparison of modeled cavity expansion limiting pressure (red) with measured earthworm abundance (black). Earthworm abundance was measured monthly in the New Forest, Hampshire UK (5.9 °N, -1.6 °E, Eggleton et al. 2009)⁶ over six consecutive years. **d-e**, Median climatic limiting pressures (bars ± IQR) required to burrow through soil are associated with mean daily precipitation⁷ (blue line and shading; 30 day running mean and SD) for **d**, a grassland (**G**: 9.6 °N, 14.7 °E) and **e**, a desert (**D**: -23.0 °N, 133.0 °E) as indicated in **a**. The maximum radial earthworm pressures P_w (dashed line) are shown. Soil limit pressures are reported for the topsoil (0-7 cm) and are assumed to represent the driest part of the soil profile. **f**, Habitat fragmentation based on habitable regions is plotted in comparison with species richness⁸ for different latitudes.

Q1.2

Instead the authors relate their predictions to observed occurrences, which is a much weaker test and I'm not left convinced that what they have done is more powerful than a more correlative approach, especially given the various additional constraints that they have added on. How much are the predictions improved as you add the various additional constraints beyond the pressure limit? Note that the activity limit should really be made to be a function of soil temperature as the speed of the earthworm life cycles will be very temperature-dependent.

A1.2

We note that the biomechanical model presented here cannot be commuted with an empirical temperature-based model. There's ample evidence as the reviewer would agree, of high temperatures and lack of earthworm activity such as in dry summer months in Mediterranean landscapes or in arid regions. The development of predictive models of natural systems is seldom accomplished in one step where all ingredients are known – this leads to incomplete and uncertain model evaluation based on available data. We have discussed in A 1.1 evidence that support the centrality of the cavity expansion pressure limit, we translated these to a spatio-temporal map informed by specific soil properties and climate to derive predictions and maps. Considering the effects of selected auxiliary constraints (*i.e.*, mean annual temperature ≥ 0 °C, sand content $\leq 80\%$, soil pH ≥ 4.5 , and Activity ≥ 2 months), we have updated Fig. 2 to include a measure of similarity (area-weighted Jaccard index) between regions classified as inhabitable considering these auxiliary constraints and by the limiting cavity expansion pressure (Fig. 2 c). Additionally, a latitudinal distribution of terrestrial area that could be excluded by each constraint is shown (Fig. 2 d):

Fig. 2: Global map of earthworm hospitable zones. **a**, Green regions indicate that annual average pressures required for cavity expansion are below the earthworm’s hydrostatic pressure limit (200 kPa). Pressures are truncated to values below 400 kPa for visualization (dark red) and permafrost regions were removed (grey). **b**, Other factors that may impede earthworm activity. Cyan regions indicate sub-zero mean annual temperature (MAT), magenta regions mark soil pH<4.5, yellow regions indicate coarse soil texture (sand content > 80%), and orange regions indicate that there are fewer than two consecutive months during which the soil mechanical properties permit cavity expansion. Regions of different limiting factors may overlap and were ordered for visibility. **c**, Overlap in area (weighted Jaccard index) that is considered hospitable based on pressure below 200 kPa (P_{200}) compared with other variables. **d**, Latitudinal distribution of terrestrial area that is excluded by considering each variable independently (colored lines) and the fully constrained habitat area (black).

The auxiliary constraints (Fig. 2 b) have been specified for completeness, while results support the hypothesis that soil hydro-mechanical conditions explain limitations to earthworm activity in most locations. We have included text to further elaborate this:

“We investigated the proportion of area where masks based on auxiliary auxiliary constraints overlap with those obtained from the cavity expansion limiting pressure P_L using Jaccard indices (Fig. 2 c). We can see that P_{200} ($P_L < 200$ kPa) overlaps with 60% of the regions limited by MAT, 90% with activity, 60% with Soil pH, and 70% with sand content. Furthermore, we highlight the latitudinal regions where the respective constraints are most influential (Fig. 2 d).

We conducted a sensitivity analyses and we can show, that considering both P_{200} and *activity* did not significantly expand the inhospitable regions. Despite only few regions where these auxiliary constraints uniquely limit earthworm activity (e.g., soil pH in the Amazon and MAT at northern latitudes), they were included to convey a more complete description of earthworm habitats. However, few mechanisms based on the additional constrains remain unresolved (e.g., low soil pH might also be a proxy for frequently flooded soils with reduced oxygenation).“

An additional comment was made regarding soil temperature effects - accounting for temperature effects on earthworm lifecycles (*i.e.*, population dynamics) and activity rates is undoubtedly important for ecological implications, however, these detailed refinements of physiological and life cycle responses are beyond the scope of this study. In principle, temperature effects could be included to refine the temporal windows for reproductive cycles (e.g., requiring longer time windows for colder temperatures). However, this would require additional assumptions and empirical links between resource availability, soil temperature and earthworm physiology. Here, we opted for a mechanistic approach, which relies on physical bounds that have general implications beyond the scope of this study (e.g., P_L also limits plant roots and has other physical and geotechnical applications). While we do not explicitly detail earthworm life cycles, we can still outline physically feasible activity windows (Fig. 4 c and d) that closely correspond to seasonal variations in population densities (see Nakamura, 1968).

Q1.3

I agree that the authors have generated a powerful index for inferring earthworm limits but they could potentially make the mechanistic aspect of their work stronger if they had a more convincing validation. The other way to approach this, especially if there are no time-series available of earthworm activity to test the model with, is to make predictions against some non-mechanistic benchmarks, e.g. that don't account for soil properties but only soil moisture. The main goal should be to more convincingly show why this approach is better than a more statistical approach, especially in regards to inferring what will happen under novel conditions.

A1.3

As the reviewer suggested, we evaluate the model (making predictions) using four independent datasets. We have more clearly specified this in the SI:

“We evaluated our model predictions with four independent datasets. Global earthworm abundance data (Johnston 2019)⁵ indicates only little abundance above the predicted pressure limit (Extended Data Fig. 4). Earthworm occurrence data (GBIF, Abbott 1994)⁵ was used to evaluate the number of true positive sites under resampling (Extended Data Fig. 6). The same data was used together with another global dataset (Phillips et al. 2019)⁸ to compare modeled thresholds to ranges of environmental conditions (Fig 2, Extended Data Fig. 5). “

Additionally, we compared the latitudinal distribution of earthworm richness (Phillips et al. 2019)⁸ to the number of habitat fragments obtained from our model (Fig 4 f). Lastly, we have also included a time series data set (Eggleton et al. 2009)⁶ of earthworm abundance for comparison with pressure dynamics (Fig. 4c). See A1.1 for details and A2.3 for comparison with non-mechanistic benchmarks.

We note that collinearity of variables might limit meaningful interpretations based on statistical inference (see Fig 2 in A1.2). Our model does not suffer from this limitation as long as input data are available and consistent. Furthermore, our mechanistic model can make predictions of several properties (occurrence, abundance and richness) whereas statistical models can typically only consider properties on which they were trained (and often model “observer density” rather than earthworm occurrence/abundance).

Q1.4

The authors could also potentially compare their predictions against date-specific observations given that they are using historical forcing data. And related to this, with regard to the monthly averaging, why not just compute the hourly movement potential and summarise those in some way (e.g. maximum runs of movement or maximum runs of no movement) rather than applying the averaging methods? This could at least be done for a subset of site to assess the effect of the monthly averaging.

A1.4

The availability of a mechanistic model makes such comparisons feasible and we thank the reviewer for pointing this out. However, we cannot assess in detail movement potential (considering an average and coarse climatic record), we can resolve exclusion (in the current form of the model using simple thresholds). We have explicitly estimated months of the year with limiting conditions (as we have done for activity). These results are not easily translated to movement potential directly. As the reviewer suggested, we have now included a time series data set that better demonstrates the prognostic potential of our mechanistic approach (Fig. 4c). See A1.1 for details. In addition, uncertainties in soil texture and water content (and/or precipitation) currently preclude meaningful descriptions of shorter

timescales (given the resolution of ~10 km at the global scale).

Q1.5

I found the discussion to be a bit long and extending beyond the analyses too much. The discussion could include more about the advantages and disadvantages of the approach taken over more correlative approaches, and cut down on the general importance of earthworms for soils (which is made in the intro).

A1.5

We have included some of the pros and cons to our approach in the context of one of the papers you've provided:

“While the mechanistic approach presented in this study requires more nuanced understanding of the underlying principles that facilitate earthworm occurrence, our methodology provides several advantages to correlative techniques. Our model can highlight where correlative models are violating causal processes and disentangle constraint collinearities, which is not feasible with correlative modeling (Kearney and Porter, 2009)⁹. As a result, our model circumvents excessive speculation that can lead to incomplete or invalid inferences (e.g. understating the importance of soil physical properties) (Phillips et al. 2019)⁸.”

Q1.6

Finally, the authors might find the following articles of interest in the context of mechanistic modelling and microclimates (there's quite a lot of other papers on this general issue of correlative and mechanistic modelling):

A1.6

Many thanks for the provided literature and all of the helpful comments. We appreciate this.

Buckley, L.B., Urban, M.C., Angilletta, M.J., Crozier, L.G., Rissler, L.J. & Sears, M.W. (2010) Can mechanism inform species' distribution models? Ecology Letters 13, 1041–1054.

Dormann, C.F., Schymanski, S.J., Cabral, J., Chuine, I., Graham, C., Hartig, F., Kearney, M.R., Morin, X., Romermann, C., Schroder, B. & Singer, A. (2012) Correlation and process in species distribution models: bridging a dichotomy. Journal of Biogeography 39, 2119–2131.

Kearney, M.R. & Porter, W.P. (2009) Mechanistic niche modelling: combining physiological and spatial data to predict species' ranges. Ecology Letters 12, 334–350.

Kearney, M.R. (2019) *microclimUS: hourly estimates of historical microclimates for the United States of America with example applications. Ecology 100, e02829.*

Kearney, M.R., Gillingham, P.K., Bramer, I., Duffy, J.P. & Maclean, I.M.D. (2020) *A method for computing hourly, historical, terrain-corrected microclimate anywhere on earth. Methods in Ecology and Evolution 11, 38–43.*

Abstract

Q1.7

line 17 - change 'promotes' to 'alters' or 'affects', or be specific about the context in which it 'promotes' functioning (e.g. agricultural)

A1.7

We've changed the sentence to specify "Earthworms activity modifies soil structure and promotes ecological and hydrological soil ecosystem functions for agricultural systems"

Q1.8

line 26 - change 'migration' to 'range expansion' or something similar unless you mean potential to get to new places rather than to persist once arrived

A1.8

We've change the text to state "...potential for earthworm migration *via* connectivity of hospitable sites..."

Main

Q1.9

line 29 - change 'sustains' to 'alters' or 'influences' or something similar

A1.9

Done.

Q1.10

line 32 - change to 'improve groundwater recharge and soil water retention, and'

A1.10

Done.

Q1.11

line 37 - add 'of' before 'up to'

A1.11

Done.

Q1.12

line 43 - earthworms' (i.e., add possessive apostrophe)

A1.12

Fixed. Thanks.

Q1.13

line 71 - put the '(iii)' after 'to'

A1.13

Done.

Q1.14

line 99 - it would be good if more information can be provided on this figure as it is such an important threshold - how sensitive are the results to variation in it?

A1.14

We've now noted this in the main text and included these details in the SI.

Results

Q1.15

- line 117 - this time window will be strongly temperature sensitive

A1.15

We've included this point in the text.

"We note that these time windows may be sensitive to temperature (e.g., lower temperatures may require longer windows of activity). However, an in-depth analysis of this is outside of the scope of this study."

Q1.16

- line 152 - is a 'climatic year' any different to a 'year'?

A1.16

We have included in the line:

"... for a climatic year (*i.e.* an ensemble year considering several decades).."

Methods

Q1.17

- generally check the tense used throughout the methods, including the supplement, and keep it past tense, e.g. the first sentence is past tense and then the next two are present tense

A1.17

Thanks for that. We've adjusted the tenses.

Q1.18

line 291 - change 'upper most' to 'uppermost'

A1.18

Done.

Q1.19

line 309 - "decelerate earthworms' metabolic cycles to critical states" is vague and the argument is not strong - I think the low temperature constraint needs some further justification/exploration

A1.19

We also stated that sub-zero temperatures would ultimately lead to earthworms freezing.

Supplement

Q1.20

- line 35 - remove second period after 'rate'

A1.20

Done.

Q1.21

- line 43 - Make this a full sentence or remove the capital L in 'leading'

A1.21

Done. Thanks.

Q1.22

- line 45 - change "earthworms hydroskeleton" to "an earthworms' hydroskeleton"

A1.22

Fixed.

Q1.23

- line 56 - change 'condition' to 'conditions' and add 'by' after 'or'

A1.23

Fixed.

Q1.24

- line 70 - "The relations of coefficient to soil fine fraction" is not a full sentence

A1.24

We've fixed the statement to be linked with the previous "i.e. the relationship with soil fine fraction".

Q1.25

- line 78 - 'fit' should be 'fitted'

A1.25

Done.

Reviewer #2 (Remarks to the Author):

The paper presents a model of the physical and other environmental limitations to earthworm activity, the former being novel and allowing new insight into earthworm ecology. The model is implemented to predict earthworm distribution at a global scale and provides an important contribution to the field. The study is generally well conducted, well written and presented with some brilliant figures. The work is original and of wide interest to the field. However, the study does fall short of some of its larger claims. These are outlined below under ‘major comments’, followed by some more ‘minor suggestions’.

We appreciate the kind and encouraging comments and hope that the amendments made in the manuscript adequately address the shortcomings of the first draft. In particular, we focused on clarifying the relative contributions of additional variables, choices of thresholds and discuss limitations to model evaluation.

Major comments:

O2.1

1) ‘Mechanistic model’: I have some issues in the claim that the model is mechanistic. While the limiting pressure has been parameterised based on some first principles (Eq. 1 & 2) the model itself determines whether earthworms will be present or not given five environmental limits. The results of each environmental limit are also not entirely clear and so it is difficult to disentangle the major controls on earthworm distribution.

A2.1

This is an important point (similar to one raised by the other reviewer - please see our response in A1.2.). We agree that threshold values (activity, sand content and soil pH) that were derived from literature data and are not necessarily mechanistic. However, an important distinction here is that our model was derived from mechanical considerations and not trained on data related to earthworms or observations thereof. Hence, our claim that such a model provides an independent estimate based on climate and soil properties alone coupled with physiological limiting pressure of Earthworms (note- such a limit may be updated as more information becomes available or applied to other burrowing organisms). We’ve included auxiliary constraints for completeness of the global picture to refine the conditions that give rise to presence or absences of earthworms (not only hydro-mechanical constraints).

Regions where auxiliary constraints dominate (other than mechanical pressure) are now illustrated in Fig 2 c and their dominance is illustrated. Only in few regions (e.g., Amazon) auxiliary constraints dominate the outcome. The hydro-mechanical constraint is based entirely on physical limitations, which have been studied in great detail (see Ruiz *et al.*, 2016, Ruiz *et al.*, 2017, and Ruiz and Or, 2018)^{3,10,11}. As such, the parameterizations of soil shear strength and shear modulus used here extend well beyond the scope of this study alone, and describe (objective) physical properties of soils with varying texture and water content. In other words, this study

does not rely on fitted parameters that are only defined in the context of this study. Rather, we are using physically measurable properties, which are more general than their application to ecological modeling.

Q2.2

2) The limiting pressure has been inferred for all earthworms, regardless of their body size – which varies massively and will be directly associated with hydrological functioning. Surely there is error associated with this assumption, but it is not mentioned.

A2.2

We agree with the reviewer that this omission should be addressed. We have included the following text in the discussion:

“We note that there are several challenges when trying to relate earthworms’ hydroskeletal pressures with earthworms’ biomass. Pressures translate to stress. Although pressures are often confounded with forces, these do not necessarily scale similarly with earthworm size. Earthworms with larger biomass are likely to exert greater forces, yet, this may not necessarily translate to larger pressures. Large anecic earthworms (*L. terrestris*) consistently demonstrated to exert lower pressures compared with smaller endogeic earthworms across several studies (McKenzie and Dexter, 1988, Quillin, 1999, and Ruiz and Or, 2018)¹⁻³. However, we are not precluding the possibility of earthworms that may exert larger pressures (e.g., large worms found in Australia). As more information becomes available, the spatial extent and constraints can be easily revised based on the mechanistic model. Furthermore, given a model sensitivity analysis on the limiting pressure (Extended Data Fig. 8), we are confident in our current model predictions.” We have compared a range of earthworm limit pressure thresholds (earthworm “strengths”) to the proportion of occurrences that lie in the designated areas:

“Extended Data Fig. 8 shows that using a threshold of 200 kPa (that was determined experimentally (McKenzie and Dexter, 1988, Quillin, 1999, and Ruiz and Or, 2018)¹⁻³) recovers almost 90% of global occurrences and indicates that the value represents an inclusive upper bound (covering the wide range of observed earthworm “strengths”). This is in agreement with earthworm abundance data from an independent study (Johnston et al, 2019⁵; Extended Data Fig. 4) where around 90% of abundances occurred below 200 kPa. Given our observations on earthworm habitat distributions and abundance timeseries (Fig 4 c, **see A1.1**), a more conservative value of 100 kPa could be used operationally to delineate regions as earthworm habitats (and would correspond to around 75% of occurrences). Here we use the more inclusive, physically-based upper bound of 200 kPa.”

Extended Data Fig. 8 Proportion of allotted occurrence as a function of earthworm’s limiting pressure. 200 kPa was considered the earthworm’s maximum pressure based on experimental measurements (McKenzie and Dexter, 1988 , Quillin, 1999, and Ruiz and Or, 2018)¹⁻³. This analysis shows that increasing the pressure threshold to 300 kPa does not increase the proportion of occurrences with the associated areas. Reducing the pressure down to 100 kPa reduces the proportion of occurrence by about 5%. Reductions in pressure below 100 kPa results in a large drop in correspondence with observations.

O2.3

3) Following on from a comparison of environmental factors on earthworm distribution in comment 2, no meaningful metrics are given for the model’s goodness of fit to the observed data. By providing such metrics, and a breakdown of the metric of each environmental factor in isolation this would be more useful in some ways than the maps.

A2.3

In the revision, we focused on improving the statistical tests and model performance metric. Unfortunately, we cannot provide metrics such as goodness of fit, because we are not fitting our model to earthworm data. For a breakdown of modeled variables see A1.2. However, relative contributions of environmental factors are difficult to infer (statistically) due to strong collinearities among variables. In addition, presence-only (positive and unlabeled) data limits meaningful model validation that would also require additional assumptions (e.g., on background sampling). Furthermore, Extended Data Fig. 6 illustrates the robustness of the true positive hit rate of the mechanistic model under variation of sample size. We further highlight these metrics in the main text.

We have included in the SI and ED subsections outlining a comparison between our mechanistic approach and two data-driven approaches for earthworm presence-only data

“Metrics that infer goodness of fit cannot be applied to our mechanistic framework, as our model is not fit to the earthworm occurrence data. In this section, we present a comparison between our mechanistic model and two data-driven approaches for earthworm presence-only data (gridded to $0.1^\circ \times 0.1^\circ$, $n = 7156$). The first method used is a one-class support vector machine (Drake et al., 2006¹²) (oc-SVM, with parameters $\eta = 0.1$ and $\gamma = 0.5$), which is an unsupervised algorithm that outputs a decision function that indicates positive and negative classifications (i.e., occurrences). The second approach is based on the Mahalanobis distance (Etherington and Thomas, 2019¹³) (D^2), which measures the distance of a point to the center of a multivariate normal distribution (a critical distance is found using chi square statistic assuming a p-value of 0.01). Thus, occurrences are classified based on values that are inliers (or “within the niche space”). Both methods have been used in ecology as they closely relate to the concept of fundamental niches. To provide a non-mechanistic benchmark, we used the following covariates (values were scaled to their range for the oc-SVM): mean annual temperature, soil pH, sand content and mean annual precipitation. We then compared the resulting masks of habitat suitability (Extended Data Fig 9).

The masks obtained from the different methods are similar and differences in true positives are smaller than the proportion of occurrences at river corridors and in certain regions in the permafrost (~10%). However, we note that while a data-driven approach may provide similar results, our mechanistic model has the added benefit of explaining the physical principles underpinning the map. As such, data-driven approaches may lead to flawed or incomplete conclusions, for example, regarding the importance of soil properties and the estimation of tropical earthworm diversity and abundance (see Phillips *et al.* 2019 with erratum and correspondences).”

Extended Data Fig 9. Masks produced by the data-driven one-class support vector machine (oc-SVM) and the Mahalanobis distance (D^2) compared to our mechanistic method. Proportion of true positives are also shown.

Q2.4

4) The abstract claims the model’s applicability to investigate how land use change alters earthworm distribution, but no analysis has been conducted to investigate this.

A2.4

We have not included analysis for this explicitly. We do discuss our results in the context of developing more sustainable land use. We have rephrased the final sentence in the abstract. “The mechanistic model delineates potential for earthworm migration and regions sensitive to climate.”

Minor suggestions:

Q2.5

L17: Earthworm

A2.5

Done.

Q2.6

L17-L20: I would suggest linking the two sentences together for a concise introduction e.g. Earthworm activity modifies soil structure and promotes hydrological ecosystem functions, but this relationship is constrained by soil hydromechanical conditions.

A2.6

In coordination with the other reviewer's comments, we have rephrased the statements to read:

"Earthworms activity modifies soil structure and promotes important ecological and hydrological soil ecosystem functions for agricultural systems. Earthworms use their flexible hydroskeleton to burrow and expand biopores. Hence, their activity is constrained by soil hydromechanical conditions that permit deformation at earthworm's maximal hydroskeletal pressure (≈ 200 kPa)."

Q2.7

L21: 'developed here'?

A2.7*

Fixed.

Q2.8

L21-22: Find this quite confusing – do you mean 'A novel biophysical model is developed here to link the biochemical limits of earthworm activity to predict bioturbation...'?

A2.8

We mean, "A novel biophysical model is used here to link the biomechanical limits of earthworm burrowing with soil moisture and texture to predict soil conditions permitting bioturbation across biomes and climate regions."

Q2.9

L22: Suggest replacing 'inject' with 'include' or 'model'.

A2.9

Done.

Q2.10

L24-25: Any metric for this?

A2.10

We've included "earthworm occurrence pattern".

Q2.11

L25-26: Why? Major environmental controls are?

A2.11

We've included, "due to soil hydromechanical status".

Q2.12

L26-27: Again, is there more information to give here? In particular regions or land use changes?

A2.12

We've rephrased to, "The mechanistic model delineates regions that exclude earthworm migration and highlight regions sensitive to climate."

Q2.13

Extended figs 1 + 2: Is there any way of relating 'silt+clay (%)' to more commonly used bulk density? Or maybe there could be a sentence in the MS to make the link clear?

A2.13

We've included a statement in the manuscript to specify that 'silt+clay' make up the fine fraction of soil, *i.e.* not sand. However, it cannot be linked to bulk density directly without information on soil packing. Nonetheless, the relation is implicitly considered by using volumetric water content where the highest values occur in soils with high porosity (and typically low bulk density).

Q2.14

Figure 1: Brilliant figure! Minor points – the box around 2b is slightly thinner than the others, and 'Soil type' in 2d is cut off slightly.

A2.14

We are humbled by your comment. We have adjusted the box in 2b to be thicker, and we've made sure 'Soil type' is not cut off.

Q2.15

Extended Fig 3 would benefit from larger text to make it readable.

A2.15

We fully agree with the comment, however, the journal specified its preference for 8 point Arial. We will make every effort to ensure that the text is legible.

Q2.16

Fig 2: L478 – purple rather than blue, and magenta rather than red? Also find the distinction between green and cyan a bit too similar and would suggest an alternative colour for one.

A2.16

Done.

Q2.17

L112-114: Seems as though these values have been plucked out the air to make the modelled distributions fit the observations – please provide references or indicate that the evidence is provided elsewhere.

A2.17

We've included references for these values.

Q2.18

L147-148: 'seasonal activity windows that are driven primarily by precipitation' – I think others might disagree with the seasonal drivers so please provide a reference for this information or indicate how you inferred it.

A2.18

We've included a reference for this statement (Kretzschmar 1991).

Q2.19

Extended Fig 7 – do all data coordinates correspond to land surfaces? It seems that some data points are in the middle of the ocean.

A2.19

All data points do correspond to longitude-latitude coordinates on land. Some small islands have measured occurrence that cannot be resolved in the figure resolution. A note was added in the figure caption.

Q2.20

Data availability – would expect the mapped predictions to be made openly available.

A2.20

We will place the data on limiting pressures and deduced earthworm habitats in a public repository for publication.

References

- 1 McKenzie, B. M. & Dexter, A. R. Radial pressures generated by the earthworm *Aporrectodea rosea*. *Biology and Fertility of Soils* **5**, 328-332 (1988).
- 2 Quillin, K. J. Kinematic scaling of locomotion by hydrostatic animals: ontogeny of peristaltic crawling by the earthworm *Lumbricus terrestris*. *Journal of Experimental Biology* **202**, 661-674 (1999).
- 3 Ruiz, S. A. & Or, D. Biomechanical limits to soil penetration by earthworms: direct measurements of hydroskeletal pressures and peristaltic motions. *Journal of The Royal Society Interface* **15**, 20180127 (2018).
- 4 Abbott, I. Distribution of the native earthworm fauna of Australia—a continent-wide perspective. *Soil Research* **32**, 117-126 (1994).
- 5 Johnston, A. S. A. *et al.* Earthworm distribution and abundance predicted by a process-based model. *Applied Soil Ecology* **84**, 112-123 (2014).
- 6 Eggleton, P., Inward, K., Smith, J., Jones, D. T. & Sherlock, E. A six year study of earthworm (*Lumbricidae*) populations in pasture woodland in southern England shows their responses to soil temperature and soil moisture. *Soil Biology and Biochemistry* **41**, 1857-1865 (2009).
- 7 Beck, H. E. *et al.* MSWEP V2 global 3-hourly 0.1° precipitation: methodology and quantitative assessment. *Bulletin of the American Meteorological Society* **100**, 473-500 (2019).
- 8 Phillips, H. R. *et al.* Global distribution of earthworm diversity. *Science* **366**, 480-485 (2019).
- 9 Kearney, M. & Porter, W. Mechanistic niche modelling: combining physiological and spatial data to predict species' ranges. *Ecology letters* **12**, 334-350 (2009).
- 10 Ruiz, S., Schymanski, S. & Or, D. Mechanics and Energetics of Soil Penetration by Earthworms and Plant Roots - Higher Burrowing Rates Cost More. *Vadose Zone Journal* **10.2136/vzj2017.01.0021**, doi:10.2136/vzj2017.01.0021 (2017).
- 11 Ruiz, S., Straub, I., Schymanski, S. J. & Or, D. Experimental Evaluation of Earthworm and Plant Root Soil Penetration--Cavity Expansion Models Using Cone Penetrometer Analogs. *Vadose Zone Journal* **15** (2016).
- 12 Drake, J. M., Randin, C. & Guisan, A. Modelling ecological niches with support vector machines. *Journal of applied ecology* **43**, 424-432 (2006).
- 13 Etherington, T. R. Mahalanobis distances and ecological niche modelling: correcting a chi-squared probability error. *PeerJ* **7**, e6678 (2019).

Reviewers' Comments:

Reviewer #1:

Remarks to the Author:

The main suggestion I had for this MS was to include either some more direct tests of the model predictions against observations of the processes being modelled (i.e. earthworm activity windows) rather than correlation with occurrence data, or to provide further analyses to show the nature of their predictions relative to other predictive approaches including correlative ones.

The authors have taken the latter approach, and have partitioned their predictions according to the different limiting factors as well as making a comparison of seasonal patterns of occurrence/activity against their predictions. I think this has strengthened the paper considerably but I would suggest that the authors make it clear in their MS that they are proposing a hypothesis that, while based on strong mechanistic reasoning, must still be tested by observation 'on the ground'. Indeed, the model provides a strong basis for such experimental work and makes clear statements on where and when to make such tests.

Specific comments:

line 67 - suggest stating 'Here we provide evidence that' instead of 'Here we show'

line 139 - replace 'We conducted a sensitivity analyses and we can show, that' with 'We conducted sensitivity analyses which showed that'

line 143 - 'However, a few mechanisms ...'

lines 221-223 - replace 'biomass' (which could be confused with the total mass of all earthworms in soil) with 'body mass' or perhaps even better use 'body size'

lines 243-262 - you could explicitly address the life cycle aspect (including temperature effects) with a Dynamic Energy Budget model and parameters exist for earthworms - see https://www.bio.vu.nl/thb/deb/deblab/add_my_pet/species_list.html and search for family Lumbricidae and you'll find that parameters are estimated for five species. There are a few examples in the literature of application of DEB theory to earthworms, the most recent being: Rakel, K. J., Preuss, T. G., & Gergs, A. (2020). Individual-based dynamic energy budget modelling of earthworm lifehistories in the context of competition. *ECOLOGICAL MODELLING*, 432, 109222. <https://doi.org/10.1016/j.ecolmodel.2020.109222>

For an algorithm to implement such a model with soil conditions in a depth/behaviorally explicit way, see also: Kearney, M. R., & Porter, W. (2019). NicheMapR – an R package for biophysical modelling: The ectotherm and Dynamic Energy Budget models. *Ecography*. <https://doi.org/10.1111/ecog.04680>

Signed: Michael Kearney

Reviewer #2:

Remarks to the Author:

The authors have sufficiently addressed this reviewer's concerns, and in doing so have improved the clarity of the model and flow of the paper. I would argue that goodness of fit metrics such as the coefficient of determination (R^2) or AIC can be calculated without fitting data to statistical models, but I do appreciate that such metrics are not as informative when the data in question is presence or absence. I now look forward to seeing this very interesting study and the set of brilliant accompanying figures published.